# Effect of Direct Oral Anticoagulants on Treatment of Geriatric Hip Fracture Patients: An Analysis of 15,099 Patients of the AltersTraumaRegister DGU^®^

**DOI:** 10.3390/medicina58030379

**Published:** 2022-03-04

**Authors:** Rene Aigner, Benjamin Buecking, Juliana Hack, Ruth Schwenzfeur, Daphne Eschbach, Jakob Einheuser, Carsten Schoeneberg, Bastian Pass, Steffen Ruchholtz, Tom Knauf

**Affiliations:** 1Center for Orthopaedics and Trauma Surgery, University Hospital Giessen and Marburg, 35039 Marburg, Germany; aignerr@med.uni-marburg.de (R.A.); hackj@med.uni-marburg.de (J.H.); eschbach@med.uni-marburg.de (D.E.); jakob.einheuser@gmail.com (J.E.); steffen.ruchholtz@uk-gm.de (S.R.); 2Department for Trauma Surgery, Klinikum Hochsauerland, 59821 Arnsberg, Germany; b.buecking@klinikum-hochsauerland.de; 3Working Committee on Geriatric Trauma Registry of the German Trauma Society, 80538 München, Germany; ruth.schwenzfeur@auc-online.de; 4Department of Orthopedic and Emergency Surgery, Alfried Krupp Hospital, 45131 Essen, Germany; carsten.schoeneberg@krupp-krankenhaus.de (C.S.); bastian.pass@krupp-krankenhaus.de (B.P.)

**Keywords:** direct oral anticoagulants, hip fracture, geriatric patient, time-to-surgery, complications

## Abstract

*Background and Objectives:* The increased use of direct oral anticoagulants (DOACs) results in an increased prevalence of DOAC treatment in hip fractures patients. However, the impact of DOAC treatment on perioperative management of hip fracture patients is limited. In this study, we describe the prevalence of DOAC treatment in a population of hip fracture patients and compare these patients with patients taking vitamin K antagonists (VKA) and patients not taking anticoagulants. *Materials and Methods:* This study is a retrospective analysis from the Registry for Geriatric Trauma (ATR-DGU). The data were collected prospectively from patients with proximal femur fractures treated between January 2016 and December 2018. Among other factors, anticoagulation was surveyed. The primary outcome parameter was time-to-surgery. Further parameters were: type of anesthesia, surgical complications, soft tissue complications, length of stay and mortality. *Results:* In total, 11% (n = 1595) of patients took DOACs at the time of fracture, whereas 9.2% (n = 1325) were on VKA therapy. During the study period, there was a shift from VKA to DOACs. The time-to-surgery of patients on DOACs and of patients on VKA was longer compared to patients who did not take any anticoagulation. No significant differences with regard to complications, type of anesthesia and mortality were found between patients on DOACs compared to VKA treatment. *Conclusion:* An increased time-to-surgery in patients taking DOACs and taking VKA compared to non-anticoagulated patients was found. This underlines the need for standardized multi-disciplinary orthopedic, hematologic and ortho-geriatric algorithms for the management of hip fracture patients under DOAC treatment. In addition, no significant differences regarding complications and mortality were found between DOAC and VKA users. This demonstrates that even in the absence of widely available antidotes, the safe management of geriatric patients under DOACs with proximal femur fractures is possible.

## 1. Introduction

Proximal femoral fractures are one of the most common fracture entities in geriatric patients. Due to demographic changes, the total number of proximal femur fractures will continue to rise. Recent estimates project a rise from 1.26 million patients worldwide in 1990 to approximately 4.5 million in 2050 [1]. Due to the high prevalence of cardiovascular and cerebrovascular concomitant diseases in geriatric patients, a relevant proportion of patients with proximal femur fractures also take anticoagulants. Recent publications have stated that about one-third of patients suffering a hip fracture take anticoagulants [2,3].

Usually, anticoagulant treatment leads to a delay in surgery, as these drugs have to be stopped or reversed prior to surgical treatment [4]. However, time-to-surgery is known to be an important predictive factor impacting morbidity and mortality in patients with hip fractures [5,6].

In recent years, much knowledge has been gained regarding the handling of Vitamin K antagonists (VKA) in patients with proximal femur fracture [7]. In addition, in contrast to direct oral anticoagulants (DOACs), the effect of VKA can be measured using standardized laboratory parameters, which are widely available. However, for the indications of non-valvular atrial fibrillation, the treatment of venous thromboembolism and the perioperative prophylaxis of venous thromboembolism in hip and knee replacement DOACs are an often-used alternative [8]. DOACs have many advantages compared to VKAs, including comparable efficacy, improved safety profile and that regular laboratory chemical monitoring is not necessary. This has led to an increase in the use of DOACs. A few recent studies have found an increase in time-to-surgery in patients receiving DOACs [9,10]. However, there is limited knowledge regarding the management of patients with proximal femur fractures taking DOACs.

Therefore, the aim of the present study was to describe the course of treatment of patients with proximal femoral fractures taking direct oral anticoagulants. In addition, time-to-surgery, complications and mortality were compared between patients taking DOACs, VKAs and patients not taking anticoagulants.

## 2. Materials and Methods

The ATR-DGU was founded in 2016 by the DGU. This multi-center database provides pseudonymized and standardized documentation of data in patients aged 70 or older with a proximal femur fracture requiring surgery.

Participating centers transmit their pseudonymized data via an Internet-based platform. Currently, about 100 hospitals from Germany, Switzerland and Austria contribute to the ATR-DGU. The scientific management is carried out by the Working Committee on Geriatric Trauma Registry of the DGU. Approval for scientific data analysis from the ATR-DGU is granted via a peer-review process in accordance with the publication guidelines. The present study is in accordance with the publication guidelines of the ATR-DGU and is registered as ATR-DGU project ID 2019-004. Data are collected in five consecutive phases: admission, pre-operative, surgery, 1st post-op week, and discharge/transfer. Furthermore, an optional follow-up can be scheduled for day 120 postoperatively. On days 7 and 120 postoperatively, health-related Quality of Life (QoL) is measured with the EQ-5D-3L questionnaire.

For the purposes of the current study, pathologic fractures of the hip, as well as periprosthetic and peri-implant hip fractures were excluded. Our analysis used the following data from patients included between 2016 and 2018: age, gender, “American Society of Anaethesiologists” (ASA)–Score, anticoagulation on admission, type of fracture, time-to-surgery; type of anesthesia, surgical complications, mortality and length of stay. All patients in hospitals contributing to the ATR-DGU received ortho-geriatric treatment.

### 2.1. Statistical Analysis

All data were summarized by frequency and percent for discrete variables and median and interquartile range (IQR) for continuous variables. Comparisons between anticoagulation groups (DOACs vs. VKA vs. no anticoagulation) were made using the Χ^2^-test for categorical variables and the Mann–Whitney U test for continuous variables.

To explore associations of time-to-surgery with therapeutic anticoagulation, Kaplan–Meier curves were estimated and compared by a two-sided log-rank test. Then, the influence of the therapeutic anticoagulation in time-to-surgery, as well as frequency of surgical complications, was evaluated in a Cox proportional hazard model or logistic regression model, and adjusted hazard ratios (HRs) or odds ratios (OR) with their respective 95% confidence intervals (CI) were estimated. Age and ASA-Score (<3 vs. ≥3) were hereby considered potential confounders. All analyses were performed using R version 3.5.2 [11].

### 2.2. Ethics

Written patient consent was obtained by participating hospitals. The data from the ATR-DGU received full approval from the Ethics Committee of the medical faculty of the Philipps-University, Marburg, Germany (AZ 46/16).

## 3. Results

### 3.1. Baseline Characteristics

In total, 15,099 patients were included in the study. Most of the patients—71.9% (n = 10,811)—were female. The median age was 85 years (IQR 80–89 years), and 77.0% (n = 11,473) of the patients had an ASA-Score ≥3. The most common fracture types were pertrochanteric (49.9%; n = 7537), followed by femoral neck fractures (45.8%; n = 6908) and subtrochanteric fracture (8.7%; n = 654). Median time-to-surgery was 17.6 h (IQR 7.1–25.8 h). In 2.9% (n = 434) of the cases, a surgical revision was necessary during the inpatient stay. These were most frequently soft tissue procedures (42.4%; n = 184), and 5.4% (n = 804) of the patients died during inpatient treatment. Further baseline characteristics are shown in Table 1. As is common in analyzing registry data, not all information was available for every patient. Additional information on how many patients were available for each category is shown in Table 1.

### 3.2. Anticoagulation at Admission

In all, 53.6% (n = 7749) of the patients took anticoagulants at the time of fracture. Acetylsalicylic acid (ASS) was the most commonly used anticoagulation agent. Overall, 11% (n = 1595) of patients took DOACs at the time of fracture, whereas 9.2% (n = 1325) were on VKA therapy.

During the study period, there was a shift from VKA to DOACs. The proportion of patients treated with VKA was higher than those being treated with DOACs in 2016. This relationship was reversed in 2018, with more patients being treated with DOACs (Figure 1).

### 3.3. Influence of Anticoagulation on Type of Anesthesia Used, Surgical Complications, Length of Stay, Mortality and Time-to-Surgery

The different types of anticoagulation (DOAC vs. VKA) showed no significant difference with regard to number of complications (*p* = 1), type of anesthesia (*p* = 1) and in- house mortality (*p* = 0.158). A slight but significant difference was found regarding the length of stay (*p* ≤ 0.001). Whereas patients on DOAC treatment stayed 17.0 d (IQR 11.1–23.0), patients that were on VKA treatment stayed 17.1 d (IQR12.1–24.0). Patients without any anticoagulation stayed 15.1 d (IQR 10.0–21.1 d) (Table 2).

As shown in Figure 2, no difference regarding the time-to-surgery was seen between patients on DOACs or VKA (patients waiting for surgery after 24 h: 54% vs. 50%). However, patients who did not take any anticoagulation underwent significantly earlier surgery, namely 25% after 24 h after admission (*p* < 0.001).

The adjusted Hazard Ratios in Table 3 show that both therapeutic anticoagulations reduce the probability of early surgery with a factor of around 0.6 compared to non-therapeutic anticoagulations, regardless of the age and ASA-Score at admission. Furthermore, in patients with VKA, the adjusted odds of receiving a surgical revision during an index stay is 1.52 higher than in patients without therapeutic anticoagulation (see Table 3).

## 4. Discussion

In the current study, an overall prevalence of 11% of patients suffering hip fractures were on DOAC treatment, whereas 9.2% were taking VKA. These numbers are in line with previous literature. The prevalence of Warfarin medication in hip fracture patients was estimated to be between 5% and 10.3% [4,12]. In a recently published retrospective cohort study, Hourston et al. reported that only 4% of their patients suffering hip fractures were anticoagulated with DOACs [10]. The reduced proportion of patients taking a DOAC compared with the present study can be explained by different study periods. In the study by Hourston et al., patients were included between October 2014 and December 2016, which was earlier than in the present study. The assumption that the time of the study enrollment is critical to the prevalence of DOAC use is supported by recent literature reporting an increasing use of DOACs [13]. The present study shows that, over time, the use of DOACs is increasing not only in the general population but also among patients with proximal femur fractures. Although more patients were taking VKA in 2016, a shift to DOACs was observed, such that more patients were taking DOACs in 2018. A 15% prevalence of anticoagulation with DOACs was recently described in a patient cohort from 2016 to 2017 among patients with proximal femur fractures [14].

Several earlier studies showed that anticoagulation is associated with a delay in surgery in hip fracture patients [10,15,16]. The current study revealed a significant difference in time to surgical fixation in patients on DOAC treatment and taking VKA when compared to those not taking any anticoagulant drugs. These findings are in accordance with the findings of other authors [10,15]. Also in line with the results of the current study, Frenkel Rutenberg et al. showed a significantly increased delay to surgery in anticoagulated patients; however, the comparison between patients under DOACs and Vitamin K antagonists revealed no significant difference [17]. In contrast to these results, a Norwegian case series by Leer-Salvesen et al. did not identify a difference in surgical delay between DOAC users and non-anticoagulated patients [14].

As simple discontinuation of Vitamin K antagonists can take up to a few days to reach an adequate anticoagulation status for surgical management, different reversal algorithms have been described [7,18,19]. For DOACs, standardized therapeutic algorithms are scarce in the literature. DOACs are eliminated renally, therefore accumulation in patients with known renal failure must be considered when planning the timepoint of surgical fixation [2]. Usually no reversal of DOAC treatment is performed as antidotes are expensive and most patients can be operated on quite early after elimination of the drug [17].

The extended time-to-surgery can also be explained by the lack of standardized tests to measure the anticoagulant effect of the drugs. The uncertainty in perioperative handling of these substances is increased by the lack of consensus on the appropriate drug-free interval until hip fracture surgery [2]. Overall, several authors advocate for early surgery even in DOAC users, as they did not find increased bleeding and transfusion rates in these patients compared to non-anticoagulated patients [14,20]

No differences regarding surgical and soft tissue complications were found in this study. Inconsistent results have been published regarding the influence of anticoagulant drugs on perioperative complications in patients suffering from hip fractures. Schütze et al. described a 3.4.-fold increased risk for intraoperative blood transfusion in patients undergoing treatment, with DOACs in patients who underwent surgical fixation of inter- or subtrochanteric fractures with a proximal femoral nail within 24 h [3]. Other authors, however, did not find increased transfusion rates in patients under DOAC treatment [14,20]. No comparison to the results of the current study is possible, as transfusion was not evaluated in this registry study. An increased rate of oozing wounds in DOAC users compared to non-anticoagulated patients was reported [14]. In line with that, Frenkel Rutenberg et al. reported more wound infections in patients under VKA treatment [17]. Some studies have shown that increased delay to surgery, which is usually present in anticoagulated patients, is a risk factor for wound complications [21,22].

In line with the results of the current study, in-hospital mortality was not increased in patients on DOAC treatment in prior literature [14,17]. Moreover, 1-year mortality was no different in a retrospective case control study comparing the outcomes after hip fracture surgery between patients receiving VKAs, patients under DOACs and non-anticoagulated patients [17].

In this study, patients taking VKAs showed a slight but statistically significant increase in hospitalization time compared to patients taking DOACs. The medical significance of this small difference has to be questioned. In contrast to the results of the current study, Hourston et al. reported no effect of anticoagulation status on lengths of stay [10]. In addition, Frenkel Rutenberg did not identify significant differences regarding the lengths of stay between patients under DOACs, Vitamin K antagonists, and non-anticoagulated patients [17]. A potential reason for this difference is the considerably higher number of patients in the present study. It is possible that the number of patients in the case series mentioned was too small to identify existing differences.

A recently published case series showed that a significantly higher percentage of DOAC users received general anesthesia than non-users. Furthermore, this study showed a significant surgical delay for patients on DOAC treatment that received neuraxial anesthesia compared to general anesthesia [14]. In this study, spinal anesthesia was performed less frequently in patients taking VKAs or DOACs compared to non-anticoagulated patients. No significant difference in the form of anesthesia was found between patients taking VKAs and patients taking DOACs. However, when comparing the data from the present study with the case series from Leer-Salvesen et al. it is remarkable that, in the case series from Norway the majority of patients (90%) were operated on under spinal anesthesia, whereas in this study, 92% of the patients were operated on under general anesthesia.

### Strengths and Limitations

First, the results of the current study represent a retrospective analysis, with potential bias. Nevertheless, data for the ATR-DGU were collected prospectively. Furthermore, the ATR-DGU does not include other relevant complications other than surgical complications, for example, nosocomial infections. Likewise, no information on intensive care unit treatment is available. Moreover, detailed information on comorbidities, renal function, laboratory parameters and prior medication was unfortunately not available due to the fact that this was a registry study. Similarly, no information on blood transfusion and anticoagulation reversal is available. The *p*-values from these tests should be interpreted with caution, because with large sample sizes, even small differences can be significant. Another weakness of the present study is that the exact time of the last intake of oral anticoagulants was not recorded. However, our study presents data from a large sample of 15,099 patients treated in ortho-geriatric settings. In addition, considering the individual years of the study period separately allows conclusions to be drawn about the varying prevalence of anticoagulant use over time. Another strength is that all patients included in the current study received ortho-geriatric treatment.

## 5. Conclusions

This large registry study showed an increased time-to-surgery in patients taking DOACs and patients under VKAs compared to non-anticoagulated patients. This underlines the need for standardized multi-disciplinary orthopedic, hematologic and ortho- geriatric algorithms for the management of hip fracture patients under DOAC treatment. In addition, no significant difference with regard to complications, type of anesthesia and mortality was found between DOAC and VKA users. This demonstrates that even in the absence of antidotes, the safe management of geriatric patients under DOACs with proximal femur fractures is possible.

## Figures and Tables

**Figure 1 medicina-58-00379-f001:**
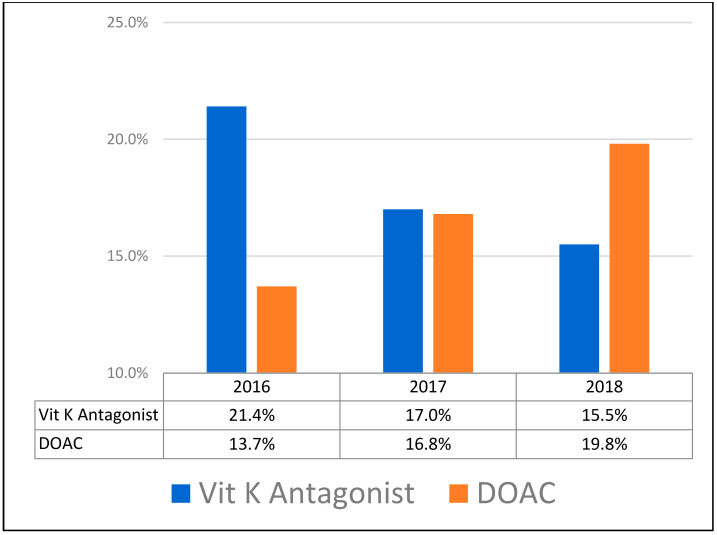
Comparison of the prevalence of Vitamin K Antagonist and DOAC treatment at admission over the study period 2016–2018 (percentage of all anticoagulated patients).

**Figure 2 medicina-58-00379-f002:**
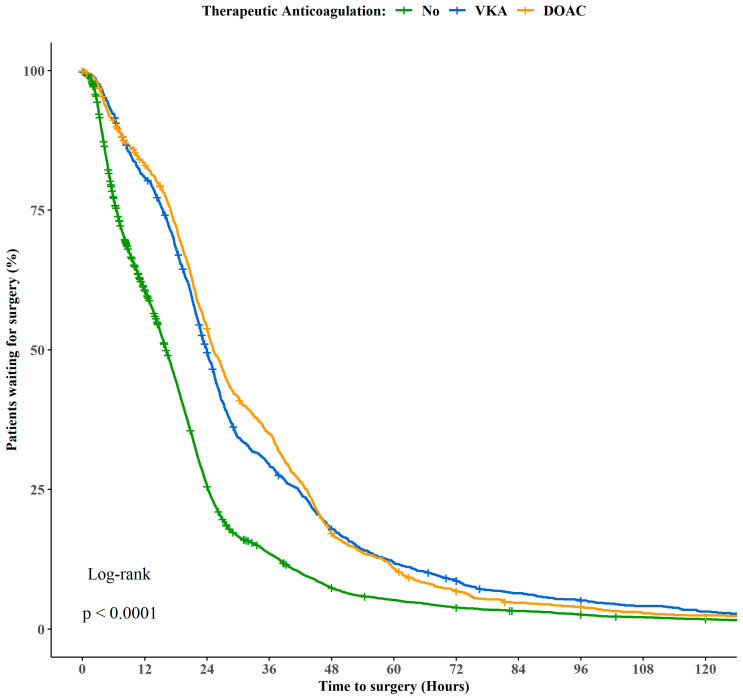
Kaplan–Meier curves of time-to-surgery for therapeutic anticoagulation treatment. VKA: Vitamin K Antagonist; DOAC: direct oral anticoagulants.

**Table 1 medicina-58-00379-t001:** Baseline characteristics of the study population.

Patient Characteristics	n = 15,099 Patients
Age (n = 14,882 patients)	85 years (80–89 years) *
Gender (n = 15,047)	
female	71.9% (n = 10,811)
ASA-Score (n = 14,898)	
1	1.4% (n = 209)
2	21.6% (n = 3216)
3	68.8% (n = 10,249)
4	8.1% (n = 1210)
5	0.1% (n = 14)
Anticoagulation on admission (n = 14,469)	
No Anticoagulation	46.4% (n = 6720)
Vitamin K antagonist	9.2% (n = 1325)
Acetylsalicylic acid	30.7% (n = 4448)
Other thrombocyte aggregation inhibitors	4.0% (n = 583)
Direct thrombin inhibitor (Dabigatran)	1.6% (n = 234)
Direct Factor Xa inhibitor (Rivaroxaban, Apixaban, Edoxaban)	9.4% (n = 1361)
Heparin	1.4% (n = 206)
Other	0.9% (n = 131)
Type of fracture (n = 15,099)	
Femoral neck fracture	n = 6908
Pertrochanteric fracture	n = 7537
Subtrochanteric fracture	n = 654
Time-to-surgery (Median/IQR) (n = 14,949)	17.6 h (7.1 h–25.8 h)
<12 h	36.4% (n = 5447)
12–24 h	34.7% (n = 5192)
24–36 h	12.6% (n = 1883)
36–48 h	7.8% (n = 1160)
>48 h	8.5% (n = 1267)
Anaesthesia (n = 14,891)	
General anesthesia	n = 13,770
Spinal anaesthesia	n = 1121
Surgical revisions (during index stay)	n = 15,080
Yes	n = 434
Reposition (after luxation)	n = 28
soft tissue intervention	n = 184
Removal of implant or osteosyntesis	n = 40
Revision of osteosynthesis	n = 62
Conversion to hemiarthroplasty	n = 25
Conversion to total hip arthroplasty	n = 30
Girdlestone	n = 5
Periosteosynthetic/Periprothetic fracture	n = 20
Others	n = 141
Mortality	
During initial stay (n = 14,944)	5.4% (n = 804)
Length of stay (Median/IQR) (n = 13,830) (survivors)	15.1 days (10.1–22.0 days)

* Median (Interquartile Range) (IQR).

**Table 2 medicina-58-00379-t002:** Comparison of anesthesia, surgical complications, soft tissue complications, length of stay and mortality during inpatient stay in patients without anticoagulation, under VKA and under DOACs.

	Anesthesia % (n = x)		Surgical Complication % (n = x)	Soft Tissue Complications % (n = x)	Length of Stay Days Median (IQR)	Mortality
	General Anesthesia	Spinal Anaesthesia				
No therapeutic Anticoagulation	91.5% (n = 10.306)	8.5% (n = 953)	2.7% (n = 304)	1.1% (n = 125)	15.1 d (10.0–21.1 d)	4.6% (n = 523)
Vitamin K antagonist	96.6% (n = 1255)	3.4% (n = 44)	4.2% (n = 55)	1.8% (n = 24)	17.1 d (12.1–24.0)	6.5% (n = 86)
DOAC	96.6% (n = 1497)	3.4% (n = 53)	3.3% (n = 52)	1.7% (n = 27)	17.0 (11.1–23.0)	8% (n = 125)
Significance VKA vs. DOAC	*p* = 1 **		*p* = 1 **	*p* = 0.951 **	*p* ≤ 0001 *	*p* = 0.158 **

* Mann–Whitney U Test, ** Chi-Quadrat Test.

**Table 3 medicina-58-00379-t003:** Adjusted Cox or logistic model to evaluate the influence of therapeutic anticoagulation on surgical-free survival or surgical complications.

	Time-to-Surgery	Surgical Complications
	HR (95%-CI)	OR (95%-CI)
Vitamin K antagonist *	0.63 (0.60–0.67)	1.52 (1.12–2.03)
DOAC *	0.61 (0.58–0.64)	1.23 (0.90–1.65)

* No therapeutic anticoagulation as reference category.

## Data Availability

Restrictions apply to the availability of these data. Data was obtained from the “Registry of the German Trauma Society” and are available from the authors on reasonable request.

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
