# Peer review of "Effect of Direct Oral Anticoagulants on Treatment of Geriatric Hip Fracture Patients: An Analysis of 15,099 Patients of the AltersTraumaRegister DGU®"

_medicina, 2022, doi:10.3390/medicina58030379_

Round 1
Reviewer 1 Report
Excellent study on outcomes of surgery in hip fracture patients showing how the rising incidence of DOACS over the use of Vitamin K analogues in Geriatric Hip Fracture patients. Good abstract with relevant introduction, appropriate Materials and Methods and accurate description of ground breaking results. Conclusion is appropriate for findings generated. Ground breaking study that is likely to reduce time to OT for DOACS. Currently in Australia majority of hospital wait 48 hrs for time to Surgery prior to OT and 72 hrs for patients with reduced creatinine clearance. The findings from this study will improve time to Surgery in our country and internationally
Author Response
Thank you for the great review of our manuscript!
Reviewer 2 Report
This is an important study and addresses new information about anticoagulants use in hip fracture patients. Suggestion is to analyse aspirin use with the same database. Some minor comments were done in the text with Track Change

Author Response
Reviewer 2:
This is an important study and addresses new information about anticoagulants use in hip fracture patients. Suggestion is to analyse aspirin use with the same database. Some minor comments were done in the text with Track Change
- Thank you very much for your great review. We adjusted the minor changes as you can see below. We focused on anticoagulants in our study and did not separately investigate antiplatelet drugs, such as acetylsalicylic acid and clopidogrel, because it would exceed the scope of the study
M1 Could be given number of non-operated hip fractures, if they appear in the registry; also statement, why numbers of patients in different patients characteristic groups are different, probably by missing data
Thank you very much for your advice. We have moved the explanation of the difference in numbers from “Material and Methods” to “Results” part for better clarity. (p. 3 l 139-141)
Only operatied patients suffering from hip fractures are available in the registry. Therefore we don’t have a number of patients that were treated without surgery.
M2 Acetetylsäure -> Acetylsalicylic acid
sThank your very much for this hint. We changed “Acetetylsäure” to “Acetylsalicylic acid” (Table 1)
This sentence could be used in conclusions like general anesthesia is safe in VKA and DOAC users, but of course this is matter of another study as well
Thank you very much for your advice. You're right, another analysis should look at the difference between regional anesthesia vs general anesthesia.
The authors thank reviewers for their helpful comments and thoughts. We are extremely thankful for the suggestions and queries and hope the additional information leads to greater clarity and understanding of the presented investigation.
Reviewer 3 Report
Effect of Direct Oral Anticoagulants on Treatment of Geriatric Hip Fracture Patients. A Retrospective Analysis of Prospectively Collected Data From 15099 Patients of the AltersTraumaRegister DGU®, by Aigner et al.
The group investigated hip fracture patients on DOCAs, derived from a large trauma registry. 11% of the HF patients were under DOACs and 9.2 % were on VKA therapy, with a strong increase towards DOACs treatment over the observation period. The median time to surgery of patients on DOACs and VKA were comparable, despite statistical significands towards earlier treatment of VKA patients (24,3 vs 23.4h) (p<0.001). Non anticoagulated Patients had a shorter time to surgeries (15.7). Complications and mortality were comparable between groups.
Major comments
# Many previous studies have reported comparable results. (Gong LN, et al Eur Rev Med Pharmacol Sci. 2021, Bruckbauer et al .J Orthop trauma 2019, Rostagno et al. Scient Reports 2021, White et al Can J Surg 2021, Gong et al Europ Rev Med Pharm Scienc 2021, ……) Therefore, the current paper lacks novelty. However, data provided by the authors fit perfect in previous scientific work. And the number of patients is high. To my knowledge, only meta-analyses provided higher numbers of patients.
# are there any data regarding comorbidities available? E.g. Carlson comorbidity index, data about renal function, eg. Creatinine or glomerular filtration rate? Or co-medication such as P-gp-inhibitors or CY3A4 inhibitors which can strongly influence the elimination half-life of DOACs?
# I would really like to see more detailed information on blood transfusion requirements, number and percentage of blood transfusion. Any difference between groups
# Is there a subgroup of patients who received high amounts of RBC transfusion? When the elimination half life is compromise due to e.g. nucleotid polymorphism (Sennesael et al. Thrombosis Journal 2018;16:28), co-medication, renal or hepatic insufficiency ect. this could result in higher bleeding rates.
# If VKA and DOACs are compared it is of importance to provide information of reversal therapy e.g. vitamin K in patients on vitamin K antagonists. Where these patients more often “reversed” compared to DOCAs. Antagonists such as PCC or Andexanet alfa are expensive and increase the risk of TE complications. Thus, these patients are not antagonized on a regular basis. (Bruckbauer et al. J Orth Trauma 2019)
# No coagulation or hemoglobin parameter upon admission and prior to surgery were provided. It is absolutely interesting if the hemoglobin levels were affected by the different antithrombotic drugs.
# Did all hospitals us standard coagulation tests in order to estimate DOAC plasma concentration. Where there some using chromogenic tests or visco-elastic testing such as ClotPro ECA-test or RVV-test (Oberladstätter et al., Anesthesia 2019, Artang TH Open 2022). If no coagulation monitoring was performed, how could the surgeons or anesthetists be sure, that patients had DOAC plasma levels lower 50ng/mL, which is considered save.
Author Response
Reviewer 3:
The group investigated hip fracture patients on DOCAs, derived from a large trauma registry. 11% of the HF patients were under DOACs and 9.2 % were on VKA therapy, with a strong increase towards DOACs treatment over the observation period. The median time to surgery of patients on DOACs and VKA were comparable, despite statistical significands towards earlier treatment of VKA patients (24,3 vs 23.4h) (p<0.001). Non anticoagulated Patients had a shorter time to surgeries (15.7). Complications and mortality were comparable between groups.
Major comments
# Many previous studies have reported comparable results. (Gong LN, et al Eur Rev Med Pharmacol Sci. 2021, Bruckbauer et al .J Orthop trauma 2019, Rostagno et al. Scient Reports 2021, White et al Can J Surg 2021, Gong et al Europ Rev Med Pharm Scienc 2021, ……) Therefore, the current paper lacks novelty. However, data provided by the authors fit perfect in previous scientific work. And the number of patients is high. To my knowledge, only meta-analyses provided higher numbers of patients.
We would like to thank the reviewer for this valuable comment and agree that similar results have been published in previous studies. However, in most cases these were small case series. The novelty of this study is the high number of patients. In addition, all patients included in the present study were treated within a standardized ortho geriatric co-management setting as described in the methods section.
# are there any data regarding comorbidities available? E.g. Carlson comorbidity index, data about renal function, eg. Creatinine or glomerular filtration rate? Or co-medication such as P-gp-inhibitors or CY3A4 inhibitors which can strongly influence the elimination half-life of DOACs?
Again, we would like to thank the reviewer for this constructive and reasonable comment. The physical health status of the patients also including the comorbidites was assessed with the ASA score. More detailed information on comorbidities, renal function, laboratory parameters, and prior medication was unfortunately not available due to the fact that this was a registry study. This was included in the limitations section.
Text change: Moreover detailed information on comorbidities, renal function, laboratory parameters, and prior medication was unfortunately not available due to the fact that this was a registry study. (p 8 l 342-345)
# I would really like to see more detailed information on blood transfusion requirements, number and percentage of blood transfusion. Any difference between groups
The authors agree with the reviewer that blood transfusion information is relevant. Unfortunately, however, this information is also not available because it was not collected in the present registry study. This has also been included in the Limitations Section.
Text change: Similarly, no information on blood transfusion and anticoagulation reversal is available. (p 8 l 342-345)
# Is there a subgroup of patients who received high amounts of RBC transfusion? When the elimination half life is compromise due to e.g. nucleotid polymorphism (Sennesael et al. Thrombosis Journal 2018;16:28), co-medication, renal or hepatic insufficiency ect. this could result in higher bleeding rates.
The authors appreciate this question. However, information on blood transfusion was unfortunately not available.
# If VKA and DOACs are compared it is of importance to provide information of reversal therapy e.g. vitamin K in patients on vitamin K antagonists. Where these patients more often “reversed” compared to DOCAs. Antagonists such as PCC or Andexanet alfa are expensive and increase the risk of TE complications. Thus, these patients are not antagonized on a regular basis. (Bruckbauer et al. J Orth Trauma 2019)
Although the authors agree that anticoagulation reversal is relevant, it must be stated again that information on anticoagulation antagonism was not collected. This was also added in the limitations section.
Text change: Similarly, no information on blood transfusion and anticoagulation reversal is available. (p 8 l 342-345)
# No coagulation or hemoglobin parameter upon admission and prior to surgery were provided. It is absolutely interesting if the hemoglobin levels were affected by the different antithrombotic drugs.
The authors appreciate this comment. However laboratory parameters were unfortunately not available in the current registry study. This was included in the limitations section.
Text change: Moreover detailed information on comorbidities, renal function, laboratory parameters, and prior medication was unfortunately not available due to the fact that this was a registry study. (p 8 l 342-345)
# Did all hospitals us standard coagulation tests in order to estimate DOAC plasma concentration. Where there some using chromogenic tests or visco-elastic testing such as ClotPro ECA-test or RVV-test (Oberladstätter et al., Anesthesia 2019, Artang TH Open 2022). If no coagulation monitoring was performed, how could the surgeons or anesthetists be sure, that patients had DOAC plasma levels lower 50ng/mL, which is considered save.
The authors appreciate this comment. However laboratory parameters were unfortunately not available in the current registry study. This was included in the limitations section.
Text change: Moreover detailed information on comorbidities, renal function, laboratory parameters, and prior medication was unfortunately not available due to the fact that this was a registry study. (p 8 l 342-345)
The authors thank reviewers for their helpful comments and thoughts. We are extremely thankful for the suggestions and queries and hope the additional information leads to greater clarity and understanding of the presented investigation.
Reviewer 4 Report
Thanks for the opportunity to review the article titled: Effect of Direct Oral Anticoagulants on Treatment of Geriatric Hip Fracture Patients. A Retrospective Analysis of Prospectively Collected Data From 15099 Patients of the AltersTraumaRegister DGU
There are some critical comments about this work:
1. The title is confusing, including both the words "retrospective" and "Prospectively" can be confusing to readers, in addition to being contradictory.
2. The most significant comment is regarding the statistical analysis. I think that the authors do not have the right approach. If my understanding of the study is correct, this research is a retrospective cohort. In which the time of surgery in three groups of patients is compared. I consider that the statistical tests do not allow the evaluation of the objectives correctly. Have the researchers tested survival analyses to measure the impact DOACs have on time to surgery?
3. The researchers used R to carry out their statistical tests; however, in material and methods, this programming language is not referenced. Authors should add a reference for R.
4. Do researchers consider that the best way to compare two continuous variables is through the Mann-Whitney U test? The sample size is huge, so the authors must assess whether this statistical test is the most appropriate.
5. In Table 1, the authors use the following notation 14.882, 15.099 and others to refer to the n values. However, this connotation is incorrect. Authors must remove the dot from the values ​​of n; otherwise, replace it with a comma.
6. Why do the authors present different values ​​of n in table 1.? Could you clarify why the value of n is sometimes 15099 and other times 14882? for example
7. I consider that the results described in point 3.3 would be more valid if they were evaluated using other types of statistical tests. See point 2.
8. In table 3, symbols are presented that are not defined
9. I consider that the calculation of OR is not the most appropriate for the design and objectives of this study.
10. It is necessary to carry out an analysis adjusted for confounding variables to measure the true impact of the use of DOACs.
Author Response
Reviewer 4:
Thanks for the opportunity to review the article titled: Effect of Direct Oral Anticoagulants on Treatment of Geriatric Hip Fracture Patients. A Retrospective Analysis of Prospectively Collected Data From 15099 Patients of the AltersTraumaRegister DGU
There are some critical comments about this work:
1. The title is confusing, including both the words "retrospective" and "Prospectively" can be confusing to readers, in addition to being contradictory.
Thank you very much for this hint. We changed the title to “: “Effect of direct oral anticoagulants on treatment of geriatric hip fracture patients. An analysis of 15,099 patients of the AltersTraumaRegister DGU®”.
The most significant comment is regarding the statistical analysis. I think that the authors do not have the right approach. If my understanding of the study is correct, this research is a retrospective cohort. In which the time of surgery in three groups of patients is compared. I consider that the statistical tests do not allow the evaluation of the objectives correctly. Have the researchers tested survival analyses to measure the impact DOACs have on time to surgery?
Thank you very much for your helpful advice, we implemented survival methods for time-to-surgery like Kaplan-Meier-Curve (Figure 2) and cox proportional hazard models (table 3)
The researchers used R to carry out their statistical tests; however, in material and methods, this programming language is not referenced. Authors should add a reference for R.
Again we would like the reviewer for this advice. We added a reference for R. (p. 3 l. 124)
Do researchers consider that the best way to compare two continuous variables is through the Mann-Whitney U test? The sample size is huge, so the authors must assess whether this statistical test is the most appropriate.
Thank you for this comment, other tests to compare two groups of continuous variables were considered but rejected due to the skewed data. That’s why instead of means and standard deviations only median and interquartile range are presented in this work.
In Table 1, the authors use the following notation 14.882, 15.099 and others to refer to the n values. However, this connotation is incorrect. Authors must remove the dot from the values ​​of n; otherwise, replace it with a comma.
Thank you very much for this advice. We added the changes to table 1.
Why do the authors present different values ​​of n in table 1.? Could you clarify why the value of n is sometimes 15099 and other times 14882? for example
- Thank you very much for your question. As common in analyzing registry data, not all informations were available for every patient.. We have moved the explanation of the difference in numbers from “Material and Methods” to “Results” part for better clarity. (p. 3 l 139-141)
I consider that the results described in point 3.3 would be more valid if they were evaluated using other types of statistical tests. See point 2.
Again we would like to thank you for this advice. We added the appropriate survival methods to the manuscript and deleted table 2 and 4.
In table 3, symbols are presented that are not defined
Thank you for this hint. We added an explanation of the symbols (Table 2)
I consider that the calculation of OR is not the most appropriate for the design and objectives of this study.
Thank you very much for this advice. For the endpoint time-to-cox proportional models with their respective Hazard Ratio were calculated.
It is necessary to carry out an analysis adjusted for confounding variables to measure the true impact of the use of DOACs
Thank you very much for this important advice. We adjusted the regression models and added it to our manuscript.
Text change: “Age and ASA-Score (<3 vs, ≥ 3) were hereby considered as potential confounders “(p.3 l.118-124. & p. 6 l.247-251).
The authors thank reviewers for their helpful comments and thoughts. We are extremely thankful for the suggestions and queries and hope the additional information leads to greater clarity and understanding of the presented investigation.
Round 2
Reviewer 3 Report
All my questions have been adequately answered. I have no more questions.
Reviewer 4 Report
No comments